# Cultural differences in social support seeking: The mediating role of empathic concern

**Shaofeng Zheng**[1]*, **Takahiko Masuda**[2], **Masahiro Matsunaga**[3], **Yasuki Noguchi**[4], **Yohsuke Ohtsubo**[5], **Hidenori Yamasue**[6], **Keiko Ishii**[1]

1 Department of Cognitive and Psychological Sciences, Graduate School of Informatics, Nagoya University, Nagoya, Aichi, Japan, 2 Department of Psychology, University of Alberta, Edmonton, Alberta, Canada, 3 Department of Health and Psychosocial Medicine, Aichi Medical University, Nagakute, Aichi, Japan, 4 Department of Psychology, Graduate School of Humanities, Kobe University, Kobe, Hyogo, Japan, 5 Department of Social Psychology, Graduate School of Humanities and Sociology, The University of Tokyo, Bunkyo, Tokyo, Japan, 6 Department of Psychiatry, Hamamatsu University School of Medicine, Hamamatsu, Shizuoka, Japan

* zheng.shaofeng@i.mbox.nagoya-u.ac.jp

**Data Availability Statement:** All relevant data are within the paper and its Supporting Information files.

## Abstract

Prior research has found that East Asians are less willing than Westerners to seek social support in times of need. What factors account for this cultural difference? Whereas previous research has examined the mediating effect of relational concern, we predicted that empathic concern, which refers to feeling sympathy and concern for people in need and varies by individuals from different cultures, would promote support seeking. We tested the prediction in two studies. In Study 1, European Canadians reported higher empathic concern and a higher frequency of support seeking, compared to the Japanese participants. As predicted, cultural differences in social support seeking were influenced by empathic concern. In Study 2, both empathic concern and relational concern mediated cultural differences in support seeking. Japanese with lower empathic concern but higher relational concern were more reluctant than European Americans to seek social support during stressful times. Finally, loneliness, which was more prevalent among the Japanese than among the European Americans, was partially explained by social support seeking.

## Introduction

Social support involves perceiving or experiencing that one is valued and cared for, is loved, and belongs to a network of communication and mutual obligation [1]. The benefits of social support to mental and physical health, such as relieving daily stress, improving well-being, and reducing the severity of health disorders [2–4], have long been known. Researchers have suggested that seeking social support from close others (e.g., family, friends) is one of the most effective ways to deal with stressful events in daily life [2, 5]. Although accepting social support helps the recipient maintain positive physical and mental health, various factors influence individuals' social support seeking. Culture, which is a collective-level phenomenon comprising both socially shared meanings and associated scripted behavioral patterns [6], is an important

**Funding:** The research was supported by Topic-Setting Program to Advance Cutting-Edge Humanities and Social Sciences Research Area Cultivation (#D-4), the Japan Society for the Promotion Science to Keiko Ishii. The funders had no role in study design, data collection and analysis, decision to publish, or preparation of the manuscript.

**Competing interests:** The authors have declared that no competing interests exist.

factor that affects support seeking. Assumptions on views of self and relationships, which are shared among individuals in a given cultural group, manifest as a set of psychological tendencies. For instance, East Asians were more reluctant to seek social support from others than Westerners [7–9].

Building on earlier findings, we examined the cultural underpinnings of using social support and the feeling of loneliness. Specifically, to address the limitations of previous research regarding what factors account for cultural differences in social support seeking, we tested whether social support would be related to individual differences in empathic concern, which reflect cultural norms about relationships and cultural practices about emotional suppression and expressivity and cognitive styles. We report findings from two studies using questionnaires consisting of attitudinal self-report scales. Note that some researchers have expressed skepticism about the cross-cultural validity of attitudinal self-report scales due to issues including translation, response bias, and reference groups, which we will return to and discuss in our general discussion. Caution, therefore, is needed in terms of the interpretation of cultural differences demonstrated in this research, because attitudinal self-report scales often fail to accurately reflect individuals' mental processes, although all the scales used in this research were reliable within a culture. In fact, disconnections between verbal reports and mental processes often occur [10]. In addition to using self-report scales, researchers have reconsidered an over-reliance of East-West differences reflecting on the dimension of independence and interdependence (or individualism and collectivism). Such a dichotomic comparison often fails to find expected cultural differences along the dimension, particularly in studies relying on attitudinal self-report scales (e.g., [11]), which suggests limited cultural sensitivity and predictive power [12]. The constructs of independence and interdependence (or individualism and collectivism) have been criticized in terms of their validity (e.g., [13]). Thus, we have to admit that this research, which uses self-report scales and relies on the West-East dichotomy, has theoretical and methodological flaws in the context of cultural psychology work. However, this research mainly aims to describe whether, and to what extent, cultures influence relationships among a set of variables and what factors account for the cultural differences—rather than just reporting what individuals think self-reflectingly about themselves. We conducted this research in line with the view of culture as a system of many elements [14].

## Culture and social support seeking

In Western cultural contexts, the self is often considered independent and separate from other people, whereas, in Eastern cultural contexts, it is viewed as being interdependent and connected to other people [15]. Specifically, in Western cultural contexts, people are encouraged to search for desirable internal traits and attributes and to express them. They are likely to share the assumption that the thoughts of each individual are unknowable in principle, unless expressed explicitly. Conversely, in Eastern cultural contexts, people are encouraged to find a meaningful position in social networks, with the emphasis being placed on social adjustment and accommodating others. At first glance, this East Asian emphasis on social networks and accommodating others could be confused with an inherent reliance on social support seeking to cope with stressful events. However, previous findings do not support this intuition.

What aspects of East Asian cultures lead people to rely less on social support for coping with stress? In previous studies, concern for the potential cost of support seeking in social relationships, which is known as relational concern, has been examined as an important factor in explaining cross-cultural differences in support seeking tendencies (e.g., [7, 16]). In Western cultural contexts, because people are likely oriented toward expressing their thoughts to achieve their goals, it is considered natural to disclose their problems and share them with

others to achieve their goals of coping with them. In contrast, social support recruits another person's resources (e.g., time) to help relieve one's own stress, which may potentially threaten the harmony established in an existing relationship. In East Asian cultural contexts, the emphasis on social networks and accommodation to others can lead people to maintain harmony within the networks and avoid matters that disrupt this harmonious relationship. Thus, it is reasonable to assume that people in East Asian cultural contexts would be more cautious about disclosing personal problems to enlist the support or assistance of others. This assumption was supported by Taylor et al. [7], who demonstrated that Asians are more concerned with how asking for others' help may negatively affect their current relationships and, thus, are more hesitant to seek social support. In addition, utilizing a hypothetical situation in which a person needs help, Miller et al. [17] conducted interviews and tested Indian, Japanese, and North American participants by asking questions about reliance on exchange norms, relationship maintenance concerns, and social support (e.g., comfort in asking for help). They demonstrated that Indians were less likely to endorse exchange norms than Japanese and North Americans, and that the cultural difference in exchange norms accounted for more positive social support outlooks in Indians than Japanese and North Americans. Additionally, when comparing the Japanese and North Americans, relationship maintenance concerns mediated the cultural differences in social support.

However, the current literature still falls short in identifying the other factors that account for those cultural differences. Given that both independence and interdependence are multifaceted concepts [18], and that people acquire a set of psychological tendencies linked to independence and interdependence through daily practices in a non-uniform manner within a culture [19], other factors related to culturally sanctioned ways of self and relationships can also contribute to cultural differences in social support seeking. Although few studies have examined this possibility, in the present research, we explored whether individual differences in empathic concern would provide an alternative explanation for cultural differences in seeking social support. We also examined whether the association between empathic concern and seeking social support further contributes to cultural differences in loneliness.

## Empathic concern unpackaging cultural differences in social support seeking

Empathic concern is an "other-oriented" affective empathy characterized by feelings of sympathy and concern for people in distress [20]. Empathic concern involves an orientation to attenuating or alleviating others' distress [21], which can be the primary motivation for helping behaviors [22]. People with high empathic concern were found to be more willing to help others in need (e.g., [23, 24]) and to devote more effort to volunteer activities (e.g., [25]).

Although many studies have examined the influence of empathic concern in regard to providing help, little research has considered the potential impact of empathic concern on asking for help. Research that has investigated the topic of empathy has found that higher empathic concern is accompanied by a stronger belief in the principle of care—that is, that people should help others in need [26, 27]. Evidence has also shown that a high endorsement of the caring principle not only encourages more prosocial behaviors [27] but also motivates individuals to seek social support [28]. Thus, high empathic concern, characterized by a strong belief in the caring principle, can lead individuals to seek social support in times of need. Indeed, prior research on coping found that empathy facilitated problem-focused coping, including support seeking (e.g., [29, 30]). Across four studies that employed three different survey panels, Sun et al. [31] consistently found that having higher levels of empathic concern was positively correlated with the frequency of seeking social support. Extending from the positive association

between empathic concern and social support seeking, the present research aimed to explore whether empathic concern could explain cultural differences in support seeking.

From the perspective of self-construal, East Asians with higher interdependent self-construal are traditionally expected to attend more to others and thus show more empathic concern (e.g., [32]). Indeed, country-level evidence from 63 countries suggests that collectivism, which is usually higher in East Asian countries, is positively associated with empathic concern [33]. However, in the majority of previous studies investigating individual differences in empathy assessed by a self-report scale (e.g., the interpersonal reactivity index), Westerners (compared to East Asians) are more likely to empathize with people in distress by exhibiting sympathy (e.g., [34]). These findings, however, require some speculation due to possible issues involving cross-cultural validity of a self-report scale, as mentioned earlier. For instance, Chung et al. [35] found that East Asian adolescents reported lower empathic concern than Western adolescents, whereas mainland Chinese university students scored lower in empathic concern assessments than German undergraduates [36, 37]. Moreover, American counselor trainees showed greater empathic concern than their Thai counterparts [38]. In addition to dispositional empathic concern, Atkins et al. [39] found that, compared to individuals of East Asian backgrounds, individuals of White British backgrounds exhibited more empathic concern while observing others suffering from social or physical pain. Why do East Asians show less empathic concern than Westerners? We speculated that the answer might lie in cultural differences in norms regarding emotional suppression and expressivity and cognitive styles.

Empathic concern involves emotional response (e.g., sympathy) toward unfortunate others. As mentioned previously, one of the prominent features of East Asian cultures is the pursuit of interpersonal harmony [15]. In many East Asian cultural contexts, people tend to value emotional suppression and emotional restraint more due to the goal of maintaining interpersonal harmony [15, 40, 41]. Prior research has found that, compared to Westerners, Japanese people evaluated their emotional events in daily life more moderately [42], and they expressed less emotion (e.g., [43]), especially when in the presence of others [44]. Low levels of empathic concern found among East Asians may be related to their emphasis on emotional suppression, which results in less emotional expressivity.

In addition to the variations in emotional suppression and expressivity, the cross-cultural differences in cognitive styles can also help explain the cultural differences in support seeking. Westerners tend to perceive every single object independently and attend to the object itself, whereas East Asians tend to consider the relations between objects and perceive the whole context unitarily (e.g., [45]). Accordingly, in empathic contexts, it might be less likely for people from East Asian cultures to take the unfortunate other's side and fully and exclusively empathize with them, without considering other situational factors (e.g., why or how the misfortune happened). Atkins [46] found that, compared to Americans, the Japanese tend to avoid taking a specific side in conflict situations, which also partially explains their lower affective empathy. Additionally, East Asians were more likely than Westerners to interpret suffering as the result of violating social norms and to perceive unfortunate people as being responsible for their suffering [47, 48]. Along these lines, in the context of empathy, East Asians, compared to Westerners, tend to attribute more responsibility to those suffering from misfortune, rather than fully siding with them—and thus show less empathic concern.

Given the positive association between empathic concern and social support seeking, lower empathic concern might also prevent Asians from seeking social support. That is, in addition to relational concern, empathic concern may also mediate cross-cultural variations in support seeking. We conducted two studies to examine this issue in detail.

## Culture, social support seeking, and loneliness

As previously noted, current literature contains abundant evidence supporting cultural differences in social support seeking tendencies; however, scant research has further investigated the psychological consequences. Thus, the current study focused on loneliness as a potential social-emotional outcome of cultural differences in social support seeking. Loneliness refers to a distressing situation in which individuals subjectively perceive deficiencies in certain social relationships [49]. In most cases, loneliness arises when individuals fail to satisfy the need for belonging and intimacy. By reminding individuals that they still have supportive relationships, social support can help individuals restore their sense of belonging [50] and, thus, reduce the feeling of loneliness [51]. Prior research has revealed that not only receiving social support but also practicing support seeking behaviors can effectively relieve the state of loneliness (e.g., [52–54]). Along these lines, a high degree of hesitancy toward seeking social support may be associated with a higher level of loneliness. If cultural differences exist in social support seeking, they may be reflected in the level of loneliness. Indeed, loneliness not only emerges as an outcome of personal experiences but also occurs as a pervasive social phenomenon within a larger context (e.g., culture; [55]). For example, previous research indicates that compared to Americans, Japanese and Chinese individuals reported greater degrees of loneliness [56–58]. Considering these factors, we explored the possibility that a high degree of hesitancy toward seeking social support might affect the level of loneliness varying across cultures.

## The current research

The main purpose of the current research was to examine whether individual differences in empathic concern as well as relational concern, which reflect differences in cultural norms about relationships and cultural practices about emotional suppression and expressivity and cognitive styles, can account for cultural differences in social support seeking tendencies. In Study 1, we examined social support seeking and empathic concern by testing Japanese and European Canadian participants. We hypothesized that European Canadians would be more likely than Japanese people to seek explicit social support, and that empathic concern, which would be higher in European Canadians than Japanese, would account for the cultural difference in social support seeking. In Study 2, we further tested whether empathic concern and relational concern mediate cultural differences in social support seeking by administering a separate survey to a sample population of Japanese and European American respondents. We anticipated that, compared to European Americans, Japanese participants would report lower levels of empathic concern, higher levels of relational concern, and lower frequencies of social support seeking. We further predicted that empathic concern and relational concern would mediate the cultural differences in social support seeking simultaneously. In addition, Study 2 also explored whether cultural differences in social support seeking are linked to cultural differences in the degree to which feelings of loneliness are experienced. We expected that Japanese participants would report higher degrees of loneliness than European American participants and that the cultural differences in loneliness could be explained by the cultural differences in empathic concern/relational concern and social support seeking.

## Study 1

### Method

**Ethics statement.**    This study was reviewed and approved by the ethics committees at Nagoya University, Kobe University, and the University of Alberta. The study participants

provided written informed consent at the beginning of the study. All responses were kept confidential.

**Participants.** A total of 407 Japanese undergraduate students participated, including students from Nagoya University, Japan (94 men and 110 women, $M_{age}$ = 19.82, $SD$ = 1.36) and from Kobe University, Japan (98 men and 105 women, $M_{age}$ = 19.70, $SD$ = 1.42). Also recruited for participation were 381 European Canadian undergraduate students from the University of Alberta, Canada (125 men, 254 women, and 2 unspecified, $M_{age}$ = 19.45, $SD$ = 2.16). The Canadian participants were prescreened based on their self-defined ethnicity. Based on cultural differences between Japanese and American individuals in their means of seeking support (elicited through four items on the Brief-COPE questionnaire) observed in Mojaverian et al. [9], we anticipated that a sample of 358 from each culture was needed to ensure 95% power to detect effect size ($d$) = 0.27. Of the 788 participants recruited, 17 did not complete all the measurements (six Japanese and 11 European Canadians), and thus, these participants were excluded, yielding a final sample size of 771 (401 Japanese and 370 European Canadians).

**Measures.** *Empathic concern*. We assessed empathic concern using the 7-item empathic concern subscale from the Interpersonal Reactivity Index (IRI; [20]). The empathic concern subscale measures an individual's general ability to feel concern and sympathy toward people suffering misfortunes. Participants were asked to rate how well each item described them using a 5-point scale ranging from *does not describe me well* (1) to *describes me very well* (5). Sample items included "*Sometimes I do not feel very sorry for other people when they are having problems*" (reverse scored), and "*I would describe myself as a pretty soft-hearted person.*" In this study, we used the Japanese translated version of the IRI developed by Himichi et al. [59], using the back-translation method, which confirmed adequate reliability and construct validity, for the Japanese participants. Cronbach's alphas were 0.80 for the Japanese sample and 0.67 for the European Canadian sample.

*Support seeking.* We assessed support seeking using the 2-item emotional support subscale and the 2-item instrumental support subscale from the Brief-COPE questionnaire [60], which is a short version of the COPE instrument [61]. Participants were asked to rate how often they tried to employ the practice or behavior described by each item using a 5-point Likert-scale ranging from *not at all* (1) to *very much* (5). Sample items included "*I try to get emotional support from others*" (emotional support) and "*I get help and advice from other people*" (instrumental support). These subscales have been used in several prior studies to examine cultural differences in social support seeking (e.g., [9]). Given the high consistencies reported for the four Brief-COPE support seeking items, we used the average scores of these four items as an indicator of support seeking. The Japanese participants were presented with the Japanese translated items developed by Mojaverian et al. [9] asking a Japanese-English bilingual to translate the original items and additional Japanese-English bilinguals to check the translated ones for accuracy. Cronbach's alpha coefficients for all four items were 0.91 for the Japanese sample and 0.91 for the European Canadian sample.

**Statistical analysis.** First, we examined the cultural differences in support seeking and empathic concern with independent sample *t*-tests. Subsequently, we estimated the Pearson correlation coefficient between empathic concern and support seeking. Finally, we used the SPSS PROCESS macro (Model 4) developed by Hayes (2013) to test empathic concern as a mediator of cultural differences in support seeking. Before conducting the mediation analysis, the scores of empathic concern and support seeking were centering by using the means across all individuals, and culture was coded as European Canadian = 1 and Japanese = 0. The indirect effect was estimated using 10,000 bootstrapping samples and presented as 95% bias-corrected confidence intervals (CI).

## Results and discussion

Consistent with previous work, the results of the independent sample t-tests showed there were significant cultural differences in support seeking ($t(769) = -2.11$, $p = .035$, Cohen's $d = 0.15$) and empathic concern ($t(769) = -7.92$, $p < .001$, Cohen's $d = 0.57$). Compared to the Japanese sample ($M = 3.24$, $SD = 1.02$), European Canadians ($M = 3.40$, $SD = 1.02$) sought social support more frequently. European Canadians ($M = 3.83$, $SD = 0.83$) also reported higher empathic concern than Japanese participants ($M = 3.40$, $SD = 0.68$). Empathic concern significantly correlated to support seeking in both the Japanese ($r = 0.24$, $p < .001$) and European Canadian ($r = 0.28$, $p < .001$) samples.

The results of the mediation model analysis indicated that the total effect of culture (Canadian = 1 and Japanese = 0) on social support seeking ($b = 0.15$, $SE = 0.07$, $t(769) = 2.11$, $p = .035$) was reduced when empathic concern was included in the model ($b = 0.00$, $SE = 0.07$, $t(768) = 0.06$, $p = .950$). The effect of culture on empathic concern was significant: $b = 0.43$, $SE = 0.05$, $t(769) = 7.92$, $p < .001$. Additionally, empathic concern positively predicted support seeking: $b = 0.35$, $SE = 0.05$, $t(768) = 7.41$, $p < .001$. More importantly, the indirect effect of empathic concern on the cultural differences in support seeking was significant: indirect effect = 0.15, $SE = 0.03$, 95% CI = [0.10, 0.21] (see Fig 1).

We also conducted another mediation model analysis with gender and age as covariates to test the robustness of this finding. The indirect effect of empathic concern remained significant, indirect effect = 0.12, $SE = 0.03$, 95% CI = [0.07, 0.18], indicating that the mediating effect of empathic concern was robust, even after controlling for the effects of gender and age. In addition, after controlling for the effect of gender and age, the mediating effects of empathic concern on the cultural differences in emotional support seeking and instrumental support seeking were both significant (S1 Table).

The cultural differences observed in social support seeking and empathic concern were consistent with those found in previous studies. Compared to the Japanese participants, the European Canadian participants reported higher degrees of empathic concern toward unfortunate others and sought social support during stressful times more frequently. More importantly, as predicted, empathic concern significantly mediated the cultural differences in social support seeking.

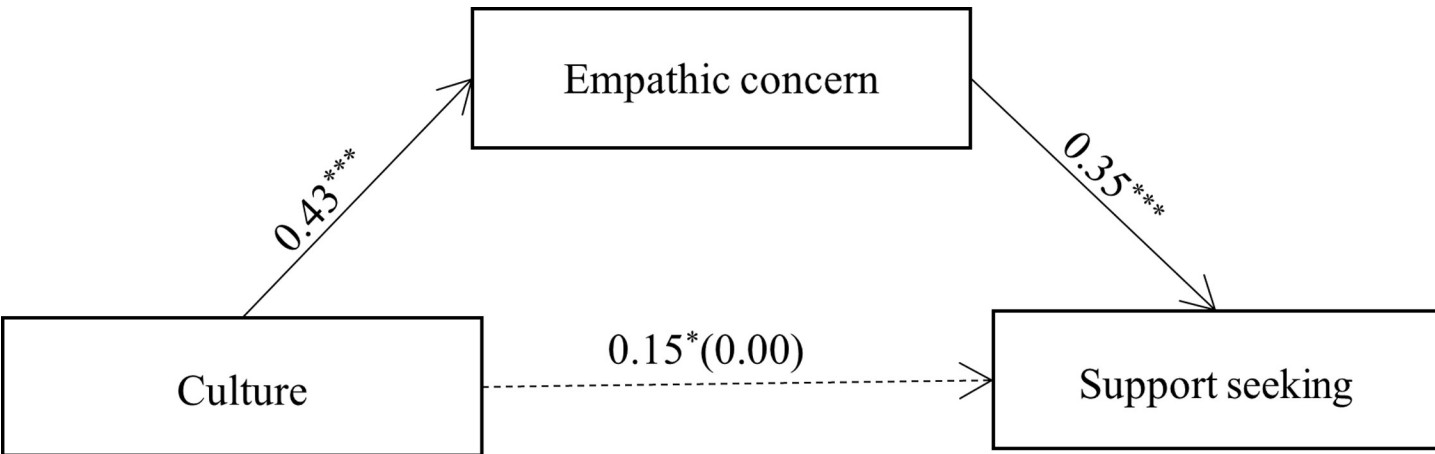

**Fig 1. Mediation model in Study 1.** Note. Culture was coded as European Canadian = 1 and Japanese = 0. $^*p < .05$, $^{***}p < .001$.

## Study 2

In Study 2, we aimed to replicate the findings of Study 1 by surveying a nonstudent sample population of Japanese and European American participants. Because most previous research has focused on relational concern in explaining cultural differences in social support seeking, in Study 2, we examined whether relational concern and empathic concern mediated the cultural differences in social support seeking simultaneously. In addition, given the association between social support seeking and loneliness [53], we also examined whether cultural differences in the degree to which loneliness is experienced would be partly due to differences in social support seeking tendencies.

We predicted the following: (a) European American participants would report higher levels of empathic concern, lower levels of relational concern, more frequent social support seeking, and less loneliness than Japanese participants; (b) social support seeking tendencies would negatively correlate with loneliness; (c) empathic concern and relational concern would both mediate the cultural differences in social support seeking; and (d) the cultural differences in loneliness would be mediated by the cultural differences in empathic concern/relational concern and social support seeking, in that order.

## Method

**Ethics statement.** This study was reviewed and approved by the ethics committee at Nagoya University. All responses were kept confidential.

**Participants and procedure.** We recruited a total of 496 Japanese participants (274 men and 222 women, $M_{age}$ = 40.29, $SD$ = 9.83) and 469 European Americans (233 men, 233 women, and three unspecified, $M_{age}$ = 38.00, $SD$ = 12.60) through online crowdsourcing marketplaces (Lancers for Japanese participants and Prolific for American participants). The American participants were recruited with filters on self-defined ethnicity (European American) and nationality (American). Based on the average effect size of Study 1 and Mojaverian et al. [9] on cultural differences in support seeking, we expected that a sample of roughly 478 for each culture would be appropriate to detect an effect size ($d$) = 0.21. Nine participants (one Japanese and eight European Americans) were excluded because they did not complete the whole questionnaire. Therefore, the final sample size was 956 (495 Japanese and 461 European Americans). After consenting, the participants completed a questionnaire used to measure stressful events, support seeking, relational concern, empathic concern, and loneliness. They were then asked to report their demographic information.

**Measures.** *Stressful events.* As performed in previous research on support seeking (e.g., [16, 62]), participants were asked to first briefly describe the biggest stressful event they had come across within the previous three months and then choose the most relevant type from nine options for their own stressors (family relationship, friend relationship, romantic relationship, academic, health, financial, job, future, or other). After recalling their stressful events, participants were asked to rate the extent to which they perceived the events as stressful, negative, solvable, and controllable and the extent to which they felt responsible for the event by responding to five statements (e.g., "*I felt responsible for this event*") using a 7-point Likert scale (1 = *not at all*, 7 = *very much*).

*Support seeking.* As in Study 1, the participants indicated how often they tried to cope with their stressors by seeking social support using 5-point scales ranging from *not at all* (1) to *very much* (5) for two emotional support items and two instrumental support items from the Brief-COPE questionnaire [60]. Cronbach's alphas for all four items were 0.86 for the Japanese sample and 0.88 for the European American sample.

*Relational concern*. Relational concern was assessed using an 11-item scale utilized in previous research (e.g., [16]). Because the current research only focused on relational concern, we did not include the items (two items) assessing the expectation of unsolicited social support in the original scale (13 items) for the main analyses. Even if the full scale (13 items) was used, the overall trends of the results remained (see S1 Text for more information). These items include several potentially negative implications of seeking support from others regarding interpersonal relationships, such as disrupting interpersonal harmony, making the problems worse, being criticized, and losing face. Sample items included "*I am concerned that if I tell the people I am close to about my problems, they would be hurt or worried for me*" and "*I would be embarrassed to share my problems with the people I am close to.*" Participants rated how important each of the concerns would be for them in deciding whether to ask for support from others using 5-point scales ranging from *not at all* (1) to *very much* (5). The items were translated and back-translated between Japanese and English by two Japanese-English bilinguals. Japanese participants were presented with the Japanese translated items. Cronbach's alphas were 0.84 for the Japanese sample and 0.92 for the European American sample.

*Empathic concern*. As in Study 1, the measurement used for empathic concern was the 7-item empathic concern subscale from the IRI [20, 23]. Participants were asked to indicate how well each statement described them. Each item was rated using a 5-point Likert-scale ranging from *does not describe me well* (1) to *describes me very well* (5). Cronbach's alphas were 0.81 for the Japanese sample and 0.88 for the European Americans.

*Loneliness*. Loneliness was assessed using the Revised UCLA Loneliness Scale (R-UCLA; [63]). The R-UCLA is a 20-item scale designed to measure the experience of social isolation and loneliness in daily life. Sample items included "*I feel in tune with the people around me*" and "*I do not feel alone*" (reverse scored). Participants were asked to indicate how often they felt the way described by the statements using a 4-point Likert-scale ranging from *never* (1) to *often* (4). Japanese participants were presented with the Japanese translated version developed by Moroi [64] that confirmed adequate reliability and construct validity. Cronbach's alpha coefficients were 0.95 for the Japanese sample and 0.95 for the European American sample.

*Demographic variables*. Participants reported their demographic information (age and gender) and their socioeconomic status (SES). SES was assessed using the MacArthur scale of subjective SES [65]. Participants were asked to look at a picture of a ladder with 10 rungs, representing the positions of people in their communities, and choose their own place on the ladder. The description of the ladder is as follows: "At the top of the ladder are the people who have the highest standing in their community (1). At the bottom are the people who have the lowest standing in their community (10)." The answers were reversely scored in the following analyses. A higher score represented a higher SES.

**Statistical analysis.**   First, following previous research, we examined cultural differences in the characteristics of stressful events. Second, as in Study 1, we examined cultural differences in the mean scores for empathic concern, relational concern, support seeking, and loneliness by conducting a series of independent sample *t*-test analyses. Then, we estimated the correlation coefficients among the variables under study.

As for the mediation analyses, we first examined the independent mediating effects of empathic concern and relational concern on the cultural differences in support seeking by conducting a mediational analysis with empathic concern and relational concern as simultaneous mediators (PROCESS v3.4 Model 4). Then, we ran a serial mediation analysis to further examine the indirect effects of cultural differences on loneliness through empathic concern/ relational concern and then support seeking (PROCESS v3.4 Model 80). All mediation analyses were conducted using Hayes's (2018) SPSS macro PROCESS v3.4, with 95% bias corrected CI based on 10,000 bootstrapping samples. As in Study 1, before the mediation analyses, the

scores of all related variables were centering and culture was coded as European American = 1 and Japanese = 0. Demographic variables and feelings related to stressful events were included in all mediation models as control variables.

## Results and discussion

**Characteristics of stressful events.** The results indicating the cultural differences in the types of stressors are depicted in Fig 2. Japanese participants were more inclined to describe stressful events related to family relationships (U.S. = 14.1%; Japan = 21.8%; $\chi2(1, N = 956)$ = 9.59, $p$ = .002) and jobs (U.S. = 21.3%; Japan = 28.3%; $\chi2(1, N = 956)$ = 6.30, $p$ = .013) than European Americans. European American participants were more likely to mention stressful events related to romantic relationships (U.S. = 5.2%; Japan = 1.4%; $\chi2(1, N = 956)$ = 10.94, $p$ = .001) and academic issues (U.S. = 4.3%; Japan = 0.8%; $\chi2(1, N = 956)$ = 12.16, $p < .001$) than Japanese participants.

In addition to the cultural differences in the types of stressors reported, Japanese participants perceived the events as more stressful ($t(881)$ = 3.56, $p < .001$, Cohen's $d$ = 0.23) and negative ($t(837)$ = 3.35, $p$ = .001, Cohen's $d$ = 0.22) than the European American participants did. Furthermore, the Japanese respondents were also more inclined to perceive the stressful events as controllable ($t(891)$ = 5.29, $p < .001$, Cohen's $d$ = 0.35) and to believe they were responsible for the event ($t(888)$ = 7.45, $p < .001$, Cohen's $d$ = 0.49). No cultural differences were observed in the participants' responses regarding their ability to resolve stressful events ($t(873)$ = 0.64, $p$ = .525, Cohen's $d$ = 0.04). Table 1 displays the mean scores by cultures.

**Cultural differences in variables under study.** The results of the independent sample $t$-tests (Table 1) showed that the European Americans tended to report higher levels of empathic concern ($t(885)$ = –9.34, $p < .001$, Cohen's $d$ = 0.61) and more frequent support seeking ($t(929)$ = –6.73, $p < .001$, Cohen's $d$ = 0.44), whereas the Japanese tended to report higher levels of relational concern ($t(847)$ = 7.44, $p < .001$, Cohen's $d$ = 0.49) and show more loneliness ($t(954)$ = 10.09, $p < .001$, Cohen's $d$ = 0.65).

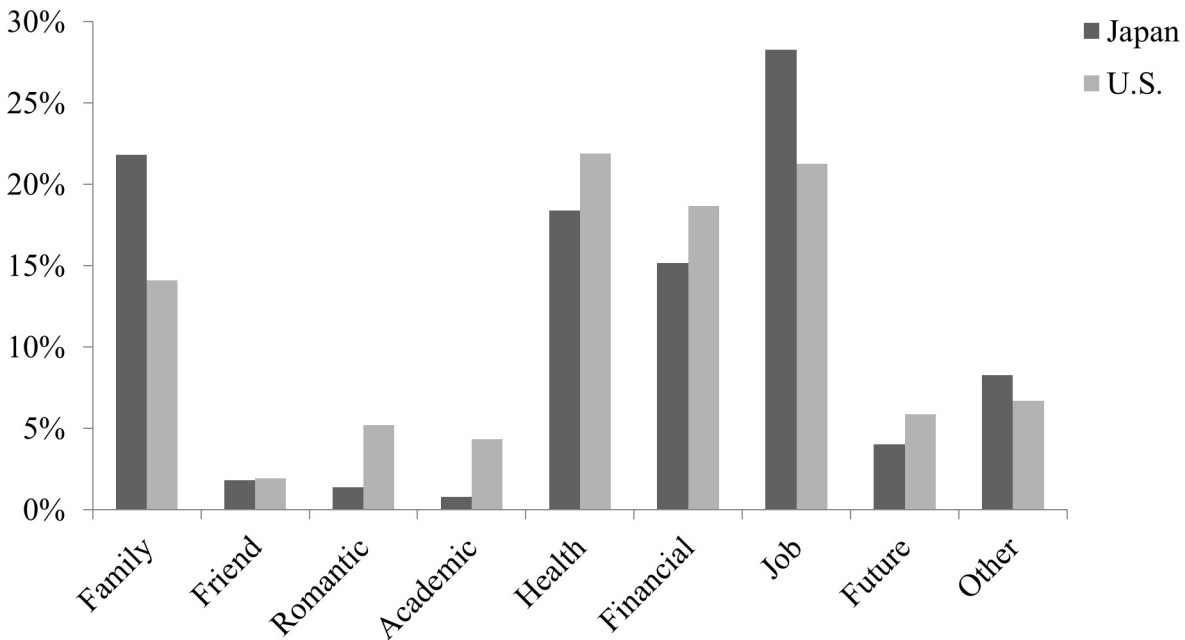

**Fig 2. Cultural differences in the types of stressors in Study 2.**

**Table 1. Means by culture in Study 2.**

| | Japanese (N = 495) | | European Americans (N = 461) | | | | | |
|---|---|---|---|---|---|---|---|---|
| | Mean | SD | Mean | SD | t | df | p | Cohen's d |
| Empathic concern | 3.31 | 0.66 | 3.76 | 0.82 | -9.34 | 885 | < .001 | 0.609 |
| Relational concern | 2.93 | 0.69 | 2.54 | 0.92 | 7.44 | 847 | < .001 | 0.486 |
| Support seeking | 2.53 | 0.94 | 2.95 | 1.03 | -6.73 | 929 | < .001 | 0.437 |
| Loneliness | 2.45 | 0.61 | 2.04 | 0.64 | 10.09 | 954 | < .001 | 0.653 |
| Stressful | 6.22 | 0.82 | 6.00 | 1.02 | 3.56 | 881 | < .001 | 0.232 |
| Negative | 5.88 | 1.19 | 5.57 | 1.64 | 3.35 | 837 | .001 | 0.220 |
| Responsible | 4.07 | 1.83 | 3.08 | 2.25 | 7.45 | 888 | < .001 | 0.486 |
| Solvable | 3.14 | 1.58 | 3.06 | 2.01 | 0.64 | 873 | .525 | 0.042 |
| Controllable | 3.20 | 1.47 | 2.64 | 1.80 | 5.29 | 891 | < .001 | 0.345 |

We conducted Study 2 in March 2020. Thus, some participants mentioned issues related to COVID-19 in their description of the stressful event. However, whether or not participants mentioned COVID-19 did not influence either support seeking or loneliness, regardless of culture (see S2 Table for more detail).

**Correlational analyses.** In both the Japanese and European American samples, empathic concern positively correlated with support seeking (Japan: $r = 0.23$, $p < .001$; U.S.: $r = 0.24$, $p < .001$), whereas relational concern negatively correlated with support seeking (Japan: $r = -0.11$, $p = .020$; U.S.: $r = -0.21$, $p < .001$). Moreover, significant negative correlations between support seeking and loneliness were observed in the Japanese ($r = -0.28$, $p < .001$) and in the European American ($r = -0.37$, $p < .001$) samples. Table 2 presents the results of the

**Table 2. Pearson correlations by cultures in Study 2.**

| | 1 | 2 | 3 | 4 | 5 | 6 | 7 | 8 | 9 | 10 | 11 | 12 |
|---|---|---|---|---|---|---|---|---|---|---|---|---|
| *Demographic variables* | | | | | | | | | | | | |
| 1 Age | - | 0.19*** | -0.10 | -0.07 | 0.05 | -0.12* | -0.11* | -0.08 | -0.20*** | 0.14** | -0.12* | -0.10* |
| 2 Gender (1 = woman, 0 = man) | -0.04 | - | 0.07 | 0.17** | 0.08 | -0.08 | -0.08 | -0.09 | -0.09 | 0.29*** | 0.07 | -0.07 |
| 3 Subjective SES | 0.03 | 0.07 | - | -0.01 | 0.01 | -0.03 | -0.01 | 0.02 | -0.09 | -0.03 | -0.00 | 0.02 |
| *Feelings for the stressful event* | | | | | | | | | | | | |
| 4 Stressful | 0.03 | 0.08 | -0.02 | - | 0.46*** | -0.11* | -0.16*** | -0.19*** | 0.06 | 0.16** | 0.25*** | -0.04 |
| 5 Negative | 0.06 | 0.01 | -0.08 | 0.44*** | - | -0.28*** | -0.36*** | -0.34*** | 0.06 | 0.10* | 0.03 | 0.12** |
| 6 Responsible | -0.07 | 0.14** | 0.02 | 0.06 | -0.12** | - | 0.37*** | 0.54*** | 0.28*** | -0.11* | -0.12** | 0.26*** |
| 7 Solvable | -0.14** | 0.05 | 0.11* | -0.10* | -0.27*** | 0.22*** | - | 0.54*** | 0.08 | -0.20*** | -0.01 | -0.04 |
| 8 Controllable | -0.07 | 0.08 | 0.12** | -0.11* | -0.29*** | 0.21*** | 0.72*** | - | 0.18*** | -0.17*** | -0.09 | 0.06 |
| *Variables under study* | | | | | | | | | | | | |
| 9 Relational concern | -0.02 | -0.04 | -0.10* | 0.00 | 0.08 | 0.03 | -0.05 | -0.04 | - | -0.25*** | -0.21*** | 0.44*** |
| 10 Empathic concern | 0.12** | 0.15** | 0.15** | 0.10* | 0.05 | 0.12** | 0.05 | 0.08 | 0.01 | - | 0.24*** | -0.26*** |
| 11 Support seeking | -0.07 | 0.17*** | 0.15** | 0.13** | 0.02 | 0.11* | 0.09* | 0.07 | -0.11* | 0.23*** | - | -0.37*** |
| 12 Loneliness | 0.01 | -0.13** | -0.35*** | 0.01 | 0.12** | -0.02 | -0.24*** | -0.22*** | 0.23*** | -0.34*** | -0.28*** | |

Note. Correlations for the Japanese sample (N = 495) are below the diagonal, and correlations for the European American sample (N = 461) are above the diagonal. $^+p = .05$,

$^*p < .05$,

$^{**}p < .01$,

$^{***}p < .001$.

correlational analyses for both samples. Given that the dependent variables were significantly correlated with demographic variables and perceived characteristics of the stressful events, we included the demographic variables and the feelings related to the stressful events (i.e., stressful, negative, responsible, solvable, and controllable) as control variables in the mediation analyses.

**Mediation analyses.**   First, we ran a multiple mediation analysis (Model 4) to examine whether empathic concern and relational concern could independently mediate cultural differences in support seeking. The results (Fig 3) showed that the effects of culture (European American = 1 and Japanese = 0) on empathic concern ($b = 0.45$, $SE = 0.05$, $t(946) = 8.89$, $p <$ .001) and on relational concern ($b = -0.25$, $SE = 0.06$, $t(946) = -4.54$, $p < .001$) were both significant. Empathic concern positively predicted support seeking ($b = 0.26$, $SE = 0.04$, $t(944) = 6.14$, $p < .001$), whereas relational concern negatively predicted support seeking ($b = -0.18$, $SE = 0.04$, $t(944) = -4.62$, $p < .001$). Moreover, the total effect of culture on support seeking ($b = 0.37$, $SE = 0.07$, $t(946) = 5.39$, $p < .001$) was reduced when empathic concern and relational concern were included in the model ($b = 0.20$, $SE = 0.07$, $t(944) = 2.96$, $p = .003$). Both empathic concern and relational concern were found to significantly mediate the cultural differences in support seeking: indirect effect = 0.12, $SE = 0.02$, 95% CI = [0.07, 0.17] for empathic concern; indirect effect = 0.05, $SE = 0.02$, 95% CI = [0.02, 0.08] for relational concern (Table 3). Additionally, the results of the mediation analyses revealed that, with 95% confidence, the difference in these two indirect effects ($d$) was significant, $d = 0.07$, $SE = 0.03$, 95% CI = [0.02, 0.13]. This indicated that the mediating effect of empathic concern was stronger than that of relational concern.

We then ran a serial mediation analysis (Model 80) to further examine whether empathic concern/relational concern and support seeking could jointly mediate the cultural differences in loneliness. The results (see Fig 3) indicated that both empathic concern ($b = -0.19$, $SE = 0.02$, $t(943) = -7.70$, $p < .001$) and support seeking ($b = -0.14$, $SE = 0.02$, $t(943) = -7.72$, $p < .001$) significantly reduced loneliness, whereas relational concern was positively associated with loneliness ($b = 0.18$, $SE = 0.02$, $t(943) = 8.19$, $p < .001$). Additionally, the total effect of culture on loneliness ($b = -0.29$, $SE = 0.04$, $t(946) = -6.69$, $p < .001$) was reduced when empathic

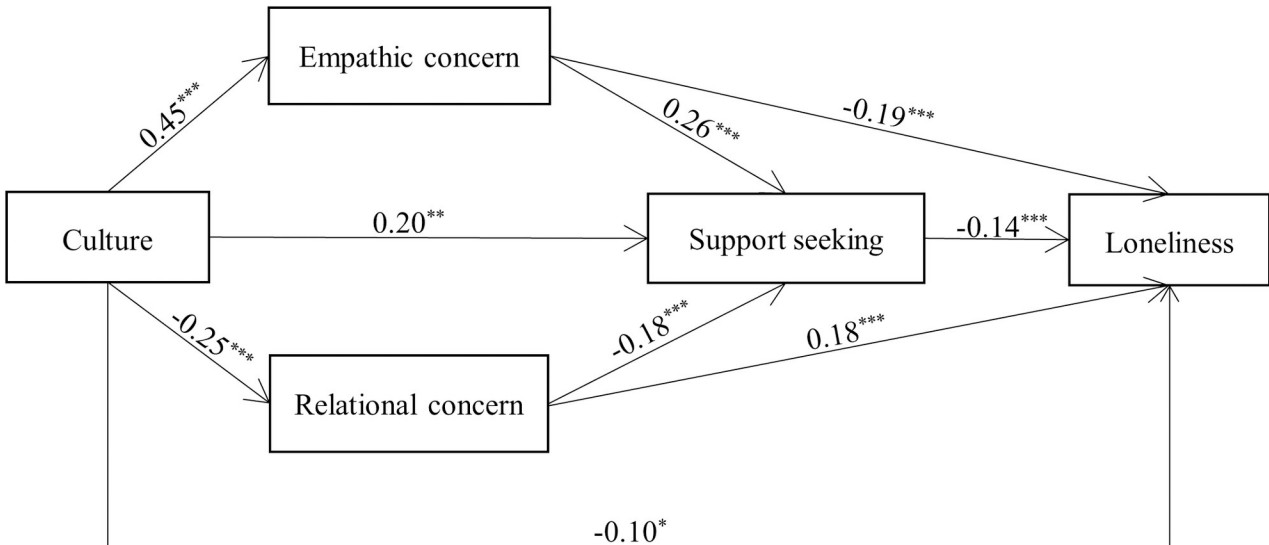

**Fig 3. Mediation model in Study 2.** Note. Culture was coded as European American = 1 and Japanese = 0. Gender, age, SES, and five related feelings for the stressful events were included as control variables. $^{*}p < .05$, $^{**}p < .01$, $^{***}p < .001$.

**Table 3. Indirect effects in Study 2.**

| | Indirect effect | SE | 95% CI |
|---|---|---|---|
| culture → empathic concern → support seeking | 0.117 | 0.025 | [0.073, 0.171] |
| culture → relational concern → support seeking | 0.045 | 0.015 | [0.021, 0.081] |
| culture → empathic concern → support seeking → loneliness | -0.017 | 0.005 | [-0.027, -0.009] |
| culture → relational concern → support seeking → loneliness | -0.007 | 0.002 | [-0.012, -0.003] |
| culture → support seeking → loneliness | -0.029 | 0.011 | [-0.053, -0.009] |
| culture → empathic concern → loneliness | -0.086 | 0.016 | [-0.120, -0.056] |
| culture → relational concern → loneliness | -0.047 | 0.013 | [-0.073, -0.024] |

concern, relational concern, and support seeking were included in the model ($b = -0.10$, $SE = 0.04$, $t(943) = -2.50$, $p = .013$). More importantly, two serial mediating effects were supported: indirect effect = $-0.02$, $SE = 0.00$, 95% CI = [$-0.03$, $-0.01$] for culture → empathic concern → support seeking → loneliness; indirect effect = $-0.01$, $SE = 0.00$, 95% CI = [$-0.01$, $-0.00$] for culture → relational concern → support seeking → loneliness (see Table 3). Furthermore, the indirect effect of cultural differences on loneliness through support seeking was also significant: indirect effect = $-0.03$, $SE = 0.01$, 95% CI = [$-0.05$, $-0.01$].

As in Study 1, we also performed the mediation analyses (Model 80, Bootstrap = 10,000) for emotional support seeking and instrumental support seeking, respectively. The results showed that all four serial mediating effects were significant (see S1 Table).

Consistent with our hypotheses, the European American participants indicated more concern for unfortunate others, less concern about the relational implication of seeking social support, a greater likelihood of seeking social support during stressful times, and less loneliness compared to the Japanese participants. In addition, more social support seeking was significantly associated with less loneliness. More importantly, Study 2 found that empathic concern and relational concern jointly mediated the cultural differences in social support seeking. Finally, consistent with our prediction, the cultural differences in empathic concern/relational concern partly explained the cultural differences in loneliness through social support seeking.

In Study 2, compared to American participants, Japanese participants perceived the stressful events they described as more stressful, negative, and controllable and felt more responsible for the events. Although we controlled the levels of participants' feelings related to the events in a series of the multiple mediation analyses, the unexpected differences in the feelings across cultures suggest that a follow-up study is warranted for the examination of the associations among empathic concern/relational concern, support seeking, and loneliness in a more controlled setting—such as a hypothetical vignette including a commonly experienced stressful event. Together, the findings of Studies 1 and 2 suggest that, in addition to higher levels of relational concern about the negative implications of seeking social support, lower empathic concern for unfortunate others also helps explain why Japanese individuals are less willing than Westerners (European Canadians and European Americans) to seek social support during stressful times.

## General discussion

Although many researchers have explained the cultural differences in social support seeking tendencies based on cultural differences in relational concern, few researchers have investigated other factors to explain the cultural differences in social support seeking. Across two studies, we found evidence to suggest that, in addition to relational concern, empathic concern also explains cultural differences in social support seeking. Through Study 1, we found that

European Canadians sought social support more frequently than Japanese individuals, when coping with stressful events, which was explained by cultural differences in the levels of empathic concern reported by the two groups. Specifically, compared to the Japanese, European Canadians were generally more concerned about unfortunate others and more willing to seek social support during stressful times. Consistent with Study 1's findings, Study 2 showed that relational concern and empathic concern mediated the cultural differences in social support seeking simultaneously. Compared to European Americans, Japanese individuals with higher levels of relational concern but lower levels of empathic concern sought social support less frequently during stressful times. The results thus replicated the previous findings and supported Study 1's initial findings. In addition, Study 2's findings revealed that Japanese participants exhibited higher loneliness than European Americans. The cultural differences in loneliness can be attributed, in part, to cultural differences in social support seeking.

Consistent with prior research, the current findings demonstrate that, compared to people from Western cultures (e.g., European Canadians and European Americans), individuals from Japan display less empathic concern for people in distress. After following the examples of previous research, we assessed the cultural difference in empathic concern by utilizing a self-report scale; however, future work should follow up the findings in a more controlled manner. Additionally, although previous research suggests that both emotional suppression and expressivity and cognitive styles might be potential candidates for explaining cultural differences in empathic concern, to date, little empirical research has directly examined the mechanisms behind these cultural differences. Future research is needed to clarify what underlies cultural differences in empathic concern.

The current work is one of the few studies that examines the effects of empathic concern on social support seeking. Considering that empathic concern is positively associated with the belief that people should help others in need [27], those who have higher levels of empathic concern may turn to others for help more naturally when they, themselves, are in distress. We also found that European Americans with higher levels of empathic concern tended to be less concerned about the negative impact of seeking support on interpersonal relationships ($r = -0.25$, $p < .001$). Recent research has shown that people with higher levels of empathic concern seek more social support for help in dealing with daily stress [31]. Our findings provide more empirical evidence supporting the positive association between empathic concern and social support seeking among multicultural samples (Japanese, European Canadians, and European Americans). These findings contribute to an understanding of individual differences in social support seeking tendencies. Social support helps people cope with daily stressful events, reduces the severity of mental and physical illness, and helps individuals adapt to new environments (e.g., [66, 67]). Thus, it is important to understand why some people are reluctant to ask for help when they are in need. Based on the current findings, it appears that possessing a low degree of empathic concern is an important factor preventing people from asking for support. Future work can further investigate whether belief in the care principle mediates the positive association between empathic concern and social support seeking.

In addition to our successful replication of the effect of relational concern, our new findings of empathic concern mediating cultural differences in social support seeking are noteworthy. Past research has suggested that, compared to Westerners, East Asians are more concerned with how explicitly enlisting support may detrimentally affect harmonious relationships and, thus, they are more reluctant to actively seek help from others [7]. Our research extends this prior work by examining the mediating effects of relational concern and empathic concern on cultural differences in social support seeking. It suggests that lower levels of sympathy for unfortunate others is another important reason why East Asians are more reluctant to seek social support than Westerners. To clarify the mediating roles of empathic concern and relational concern, future work will need to address the possibility that the underlying effects

would depend on the forms of social support seeking. Whereas this research demonstrated that empathic concern and relational concern mediated the cultural difference in explicit forms of social support seeking such as getting emotional support and advice from other people, it is unclear whether the mediating effects can be applied to more implicit forms of support seeking—defined as the emotional comfort experienced without disclosing one's problems and stress. Asians and Asian Americans likely benefit from implicit support seeking [68], and those who tend to endorse adjustment goals are likely to emphasize relational concern as a motivating factor in deciding to seek implicit social support [62]. These previous findings imply a positive association between relational concern and implicit support seeking. In contrast, does empathic concern lead people to seek implicit support as well as explicit support when they have to cope with stressful events? Future work is necessary to address this question and expound upon our findings.

This research represents one of the first scholarly efforts to examine possible social-emotional outcomes of the cultural differences in social support seeking. Our findings demonstrate that active social support seeking behaviors are effective in relieving loneliness [53], and that cultural differences in feelings of loneliness are partly due to cultural differences in social support seeking tendencies. Specifically, compared to European Americans, Japanese individuals with higher levels of relational concern but lower levels of empathic concern were more hesitant to seek social support in stressful times and, thus, suffered more loneliness. Accumulating evidence suggests that loneliness can trigger adverse outcomes on mental and physical health, such as depression [69] and alcoholism [70]. Moreover, both suicide ideation and incidents of parasuicide increased with the levels of subjective loneliness [71]. Loneliness is one of the highest risk factors for mortality [72]. Given that unsolicited support is not always available, it is important to discuss how to encourage people to seek social support actively when they are in need. For instance, it might be useful to encourage Japanese individuals who are motivated to maintain positive relationships—and, thus, have relationship maintenance concerns—to build a communal system where they can receive social support without any relational concerns. Additionally, according to our findings, encouraging people to express sympathy for unfortunate others and helping to build a more caring environment can help reduce people's hesitation to ask for social support.

In this research, we offer a new explanation for cultural differences in social support seeking. However, several limitations should be addressed. First, the present research used a cross-sectional design. Thus, we cannot exclude the possibility that higher levels of loneliness tend to cause Japanese people to cope with stress alone. Although the positive effects of support seeking on relieving loneliness have been proven repeatedly (e.g., [73]), longitudinal research is needed to further elucidate the association between support seeking and loneliness. Second, our findings lend support to prior work suggesting that East Asians have lower sympathy for people in distress than Westerners. However, we did not examine the reason for the cultural differences in empathic concern. Although we proposed possible explanations for these differences, future research is necessary to further examine the psychological mechanisms behind cultural differences in empathic concern. Third, because we only used Japanese participants, this may raise an issue regarding generalizability. There is no doubt that Japanese people cannot represent all East Asians. Miller et al. [17] suggested that the assumption that relational concern results in hesitation in social support seeking would be supported in the case that people are likely to follow exchange norms based on costs and benefits in terms of relationships with other people, including friends and siblings. Given that collectivism is positively associated with communal norms [11], however, the assumption may be exceptional; rather it may be applied to a limited group of East Asians. To gain a more integrated understanding, future work is needed to test the assumption at various sites in Asian cultures. Through comparisons among Indian, Japanese, and North American participants, Miller et al. [17] effectively

demonstrated that Indians have a more positive outlook on social support seeking than Japanese, reflecting differences in the extent to which they rely on communal norms and exchange norms. The method of triangulation [74] to identify explanatory factors of cultures by comparing subgroups will enable researchers to provide an advanced view of how the mind is shaped by cultural content beyond the dichotomy of individualism and collectivism. Additionally, given regional variations based on the history of voluntary settlement [75] and socio-ecological variations based on relational mobility [76] and residential mobility [77] in the single national culture of Japan, which can lead to differences in the extent to which individuals adhere to the dominant value of interdependence, further work is needed to examine variations in the use of social support within the specific nation. By doing so, we can specify the ways of social support seeking that emerge as adaptations to norms about relationships and emotional suppression and expressivity shared and assumed among individuals in a given sociocultural context.

Finally, although the current findings provided some preliminary evidence for the role of empathic concern in cross-cultural differences in support seeking, they were based on the usage of self-report Likert scales. Mean-level cross-cultural comparisons can be problematic for several reasons. For instance, the effects of culturally biased ideas participants rely on [13] are inescapable regardless of the careful implementation of translation and back-translation. For instance, when "my own opinion" and "directly" are used, there is no absolute standard about the concepts of these words across various cultures. Rather, participants interpret these words based on their own ideas acquired through their daily experiences in a given culture. This suggests that participants' ratings for items reflect the ideas based on their experiences, which should vary across cultures and thus are not comparable cross-culturally [78]. Additionally, ambiguity in the meaning of words used in an item often cause participants' evaluations relative to the feature of the group the participants belong to, which is called the reference group effect [79]. Due to the relative judgment by participants who use the feature of their group as a standard, self-report Likert scales have little predictable validity. Furthermore, people often present themselves in a socially desirable way when being asked about themselves. The motive for social desirability causes a response bias. Previous research found that cultural orientations were associated with tendencies to respond to questions in a socially desirable way: individualism was associated with self-deceptive enhancement, whereas collectivism was associated with impression management based on normative responses in a given culture [80]. The previous finding suggests that Westerners' higher scores for social support seeking and empathic concern and their lower level of loneliness may result from their positive views of themselves. In contrast, Japanese' lower scores of support seeking may reflect their expected normative responses, not their actual attitudes. Although the cross-cultural differences in each variable we used have been examined repeatedly with various methods (e.g., [7, 39]), the use of Likert-style questionnaires does not allow us to draw any strong inferences. Given the drawbacks of mean-level cross-cultural comparisons on Likert scales, future research is necessary to further replicate these findings by using various measurements and experimental designs.

In short, with both of our studies, we identified the mediating role of empathic concern in the cultural differences regarding social support seeking. In light of our findings, compared to European Canadians/Americans, lower empathic concern and higher relational concern discourage Japanese people from seeking social support to help cope with stress, and this reduced support seeking and worsened feelings of loneliness.

## Supporting information

**S1 Appendix. Scales used in this research.**
(PDF)

**S1 Table. Results of emotional support seeking and instrumental support seeking.**
(PDF)

**S2 Table. Means of support seeking and loneliness by COVID-19 in Study 2.**
(PDF)

**S1 Text. Results using the whole scale of relational concern (13 items).**
(PDF)

**S1 Data.**
(XLSX)

**S2 Data.**
(XLSX)

## Acknowledgments

We thank Amy Chan, Elsie Chang, Lili Gang, Miho Iwasaki, Mindy Jiang, Naoki Konishi, Maki Oba, Misaki Ochi, Angelica Paras, Shunta Sasaki, and Mana Yamaguchi for their support in carrying out this work.

## Author Contributions

**Conceptualization:** Shaofeng Zheng, Takahiko Masuda, Masahiro Matsunaga, Yasuki Noguchi, Yohsuke Ohtsubo, Hidenori Yamasue, Keiko Ishii.

**Data curation:** Shaofeng Zheng, Takahiko Masuda, Yohsuke Ohtsubo, Keiko Ishii.

**Formal analysis:** Shaofeng Zheng, Masahiro Matsunaga.

**Funding acquisition:** Keiko Ishii.

**Writing – original draft:** Shaofeng Zheng, Keiko Ishii.

**Writing – review & editing:** Shaofeng Zheng, Takahiko Masuda, Masahiro Matsunaga, Yasuki Noguchi, Yohsuke Ohtsubo, Hidenori Yamasue, Keiko Ishii.

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
