## [Decision Letter · Decision Letter 0]

9 Apr 2021

PONE-D-20-40124

Cultural Differences in Social Support Seeking: The Mediating Role of Empathic Concern

PLOS ONE

Dear Dr. Zheng,

Thank you for submitting your manuscript to PLOS ONE. After careful consideration, we feel that it has merit but does not fully meet PLOS ONE’s publication criteria as it currently stands. Therefore, we invite you to submit a revised version of the manuscript that addresses the points raised during the review process.

We look forward to receiving your revised manuscript.

Kind regards,

Frantisek Sudzina

Academic Editor

PLOS ONE

Journal Requirements:

2. Please change "female” or "male" to "woman” or "man" as appropriate, when used as a noun (see for instance https://apastyle.apa.org/style-grammar-guidelines/bias-free-language/gender).

3. We note you have included a table to which you do not refer in the text of your manuscript. Please ensure that you refer to Table 4 in your text; if accepted, production will need this reference to link the reader to the Table.

Reviewers' comments:

Reviewer's Responses to Questions

**Comments to the Author**

1. Is the manuscript technically sound, and do the data support the conclusions?

Reviewer #1: Partly

Reviewer #2: Partly

2. Has the statistical analysis been performed appropriately and rigorously? 

Reviewer #1: I Don't Know

Reviewer #2: I Don't Know

3. Have the authors made all data underlying the findings in their manuscript fully available?

Reviewer #1: Yes

Reviewer #2: Yes

4. Is the manuscript presented in an intelligible fashion and written in standard English?

Reviewer #1: Yes

Reviewer #2: Yes

5. Review Comments to the Author

Reviewer #1: Zheng et al, Social support seeking

This is an interesting set of studies on an extremely important topic, with data from three nations (I wouldn’t call them three ‘cultures’). The studies are generally well conceived and well analyzed.

However, I recommend that the paper be rewritten. First, while the authors’ summary of the the health effects of loneliness vs. social support are valid, their sources are out of date and exclude a number of important meta-analyses. That is easy to fix; I list three important sources below, but there are others as well.

Second, like many others, the authors frame their conceptualization and design in terms of the constructs of individualism and collectivism. Research shows that these are at best orthogonal dimensions, and certainly not dichotomous discrete categories. Indeed, meta-analyses strongly indicate that neither construct is valid. For one critique, see Fiske (2002) and the other comments in that issue.

Third, any time one reports studies using translations of scales, one needs to report the precise items in each language, then discuss and carefully consider the implications for data analyses of the differences in meanings of the items in the respective languages.

Finally while many researchers infer differences between nations (or cultures) by comparing means on Likert scales, such inferences are invalid. There are several reasons for this invalidity, including the fact that one cannot meaningfully compare the means of items written in different languages. Respondents in different cultures are also likely to anchor their responses differently – e.g., basing their responses on comparison of their beliefs about themselves with their beliefs about others in their own culture. Making meaningful comparisons between nations (or cultures) requires systematic ethnological research based on analyses of ethnographies, as well as consultation with social scientists who study both of the two cultures. At a bare minimum, one needs to use scenario items (Peng, Nisbett & Wong 1997; Heine, Lehman, Peng, & Greenholtz 2002). In short, I recommend that the authors not use previous studies or their own data to infer mean international differences in any of their variables.

So I strongly urge the authors to reframe their conceptualization and analyses as replications in three nations showing that social support seeking is affected by relational concerns, mediated by empathic concern and “social norms.” And I encourage a deeper conceptualization of “social norms” and how they operate in this case. This would make a solid and valuable article.

Barth, J, Schneider, S, & von Kanel, R. 2010. Lack of Social Support in the Etiology and the Prognosis of Coronary Heart Disease: A Systematic Review and Meta–analysis. Psychosomatic Medicine 72:229–238.

Chida, Yoichi, Mark Hamer, Jane Wardle, & Andrew Steptoe, J. 2008. Do Stress–Related Psychosocial Factors Contribute to Cancer Incidence and Survival? Nature Clinical Practice Oncology 5:466–475.

Fiske, A. P. 2002. Using Individualism and Collectivism to Compare Cultures: A Critique of the Validity and Measurement of the Constructs: Comment on Oyserman et al.. Psychological Bulletin, 128, 78-88. Reprinted in Deborah Cai, Ed., 2010, Intercultural Communication. London: Sage Publications.

Heine, S. J., Lehman, D. R., Peng, K., & Greenholtz, J. 2002. What's wrong with cross-cultural comparisons of subjective Likert scales?: The reference-group effect. Journal of personality and social psychology, 82(6), 903-918.

Holt-Lunstad J, Smith TB, Baker M, Harris T, Stephenson D. Loneliness and social isolation as risk factors for mortality: a meta-analytic review. 2015 PerspectPsychol Sci. Mar;10(2):227-37.

Peng, K., Nisbett, R. E., & Wong, N. Y. 1997. Validity problems comparing values across cultures and possible solutions. Psychological methods, 2(4), 329-344.

Reviewer #2: This paper examines the effects of culture on empathic and relational concern, and the combined effects of these two measures on support seeking behaviors. The authors conduct two survey studies to address these issues, and on the whole the results support their hypothesis that empathic concern is an important factor in support seeking (and outcomes that come from lack of support seeking, such as loneliness).

The framing of the paper could use improvement. First, it is important to be specific about the cultures being investigated. The literature review moves back and forth between different conceptualizations, the connections among which are not clear. For example, are the phenomena in question specific to Japan? To all East Asian countries? Do they also include those in other countries of East Asian descent, such as Chinese Americans? Similarly, the comparison groups vary widely, including Canadians of European descent, Americans of British descent, etc. (The latter is not the majority of U.S. citizens, so this comparison is not particularly relevant to understanding country level differences in help-seeking.) Care should also be taken not to use “Canadians” or “Americans” when what is meant is a subgroup of the population.

The underlying arguments regarding culture and empathic concern are quite interesting, and as the authors note, some of the prior findings would appear on the surface to be contradictory. However, the line of argument is likely to be hard to follow without a more general introduction to the cultural attributes of the cultures under comparison (e.g., individualism/collectivism, self-concept, etc.). This could make subsequent discussion of possible reasons for cultural differences in help seeking easier to follow, particularly for readers who do not work in this area of research.

The paper would also be easier to follow if all key terms and phrases were defined on first use, including “empathic concern” in the first paragraph and abstract; “implicit social support” in paragraph 2, and so on.

The two studies are entirely survey based, using adequate sample sizes of 407 Japanese and 3 European Canadians in Study 1 and 496 Japanese and 469 European Americans in Study 2. Average age of participants was around 20 for Study 1 and closer to 40 for Study 2.

The survey analysis is robust, with one limitation discussed below. The findings support the hypotheses and provide some new insights into cultural difference in empathic concern, help-seeking, and outcomes such as loneliness.

My main concern is that comparing surveys distributed to different cultural groups is tricky and the authors do not seem to have considered cultural differences in the ways in which people respond to subjective scales. There is a large literature showing that simply translating a scale between languages does not ensure that the respondents are conceptualizing the different values on the scale in the same way. There are also related issues pertaining to social desirability in responses, which also varies across cultures.

In the literature, there are several recommended solutions to this problem of comparability of response samples, including centering, testing patterns of results within samples, and so forth. Janet Harkness’ book on Cross-Cultural Survey Methods provides a good overview of the problems that arise in this kind of research and possible ways to handle them. I would strongly urge the authors to address this problem in future versions of this paper.

6. PLOS authors have the option to publish the peer review history of their article (what does this mean?). If published, this will include your full peer review and any attached files.

Reviewer #1: **Yes: **Alan Page Fiske

Reviewer #2: No

---

## [Author Response · Author response to Decision Letter 0]

21 May 2021

Editor comments:

Journal Requirements

Following the instruction on the journal webpage, we reformatted our manuscript.

2. Please change "female" or "male" to "woman" or "man" as appropriate, when used as a noun.

Following this suggestion, we revised the participants section of each study (lines 200-203 on page 10 and lines 299-300 on page 14) and Table 2.

3. We note you have included a table to which you do not refer in the text of your manuscript. Please ensure that you refer to Table 4 in your text.

This revised manuscript does not either include Table 4 or mention it in the main text.

4. Please include captions for your Supporting Information files at the end of your manuscript, and update any in-text citations to match accordingly.

Following this suggestion, we included captions for our supporting information files at the end of our manuscript and updated in-text citations. 

Reviewer 1

However,

1）First, while the authors' summary of the health effects of loneliness vs. social support are valid, their sources are out of date and exclude a number of important meta-analyses. That is easy to fix; I list three important sources below, but there are others as well.

Thank you for your suggestion. We added some new reviews/ meta-analyses of the health effects of social support and loneliness (Barth et al. [2010]; Chida et al., [2008]; Chu et al., [2010]; Holt-Lunstad et al., [2015]), which include ones you suggested (see [2], [3], [4], and [68] in the main text).

2) Second, like many others, the authors frame their conceptualization and design in terms of the constructs of individualism and collectivism. Research shows that these are at best orthogonal dimensions, and certainly not dichotomous discrete categories. Indeed, meta-analyses strongly indicate that neither construct is valid. For one critique, see Fiske (2002) and the other comments in that issue.

Previous research of culture and social support assumes the negative implications of social support seeking for interpersonal relationships, such as bothering others and losing face, in cultures (or nations) emphasizing social networks and accommodation to others. Particularly, disclosing problems and bringing personal matters to others are relatively inappropriate in Eastern cultures, because such a disclosure implies a demand for help from others, which may damage a harmonious relationship. This assumption is thus based on the norm of interpersonal relationships reflecting the interdependent orientation. On the other hand, according to Miller, Akiyama, and Kapadia (2017, Journal of Personality and Social Psychology), hesitation in seeking social support, reflecting relational concern on disrupting harmonious relationships, seems to be inconsistent with communal norms that have been considered as a core feature of collectivism (Oyserman et al., 2002). Miller et al. (2017) thus call into question two claims: one is that collectivism is associated with hesitation in seeking social support, and the other one is that the feature of hesitation in seeking social support, reflecting the interdependent orientation, is generalized in Asians (indeed, they found Indians who likely to engage in communal norms less hesitate social support seeking compared to Japanese). Given your comment and the findings of Miller et al. (2017), we reframed the introduction of our manuscript along with the previous research on culture and social support, not mentioning the dimension of individualism-collectivism (on pages 3-5). Additionally, based on Miller et al. (2017), we mentioned a possibility that the association between relational concern and hesitation in social support seeking may be applied to a limited group of East Asians on page 28. 

3) Third, any time one reports studies using translations of scales, one needs to report the precise items in each language, then discuss and carefully consider the implications for data analyses of the differences in meanings of the items in the respective languages.

Based on this comment, we included the English and Japanese versions of all the scales we used in Supporting Information (S1 Appendix) to make them public so that people who are interested in the scales can evaluate the validity. Also, we added some words to support cross-cultural validity for each scale (on pages 11, 16 and 17). Furthermore, we mentioned an issue of culturally biased item wordings by referring tot Fiske (2002) on page 28.

4) Finally while many researchers infer differences between nations (or cultures) by comparing means on Likert scales, such inferences are invalid. There are several reasons for this invalidity, including the fact that one cannot meaningfully compare the means of items written in different languages. Respondents in different cultures are also likely to anchor their responses differently - e.g., basing their responses on comparison of their beliefs about themselves with their beliefs about others in their own culture. Making meaningful comparisons between nations (or cultures) requires systematic ethnological research based on analyses of ethnographies, as well as consultation with social scientists who study both of the two cultures. At a bare minimum, one needs to use scenario items (Peng, Nisbett & Wong 1997; Heine, Lehman, Peng, & Greenholtz 2002). In short, I recommend that the authors not use previous studies or their own data to infer mean international differences in any of their variables.

As you pointed out, we admit that the problem of mean-level cross-cultural comparison using Likert scales is one of the limitations of this research. We mentioned the problem by referring to Heine et al. (2002) and Peng et al. (1997) on page 28. However, please note that our hypothesis is not based on cultural differences in traits and thoughts themselves people support. Rather, we focus on how a set of variables related to relationships and emotional experiences are functionally related and whether culture moderates the functional relationships. Following Kitayama (2002), which is one of the comments to Oyserman et al. (2002), we take a stand for the system view of culture. 

So I strongly urge the authors to reframe their conceptualization and analyses as replications in three nations showing that social support seeking is affected by relational concerns, mediated by empathic concern and "social norms." And I encourage a deeper conceptualization of "social norms" and how they operate in this case. This would make a solid and valuable article.

In this revision, reframing the introduction of culture and social support seeking by addressing the assumption of the previous research, we emphasized the roles of relational concern and empathic concern as mediators. For example, we mentioned that the results (i.e., compared to European Americans, Japanese individuals with higher levels of relational concern less frequently sought social support during stressful times) replicated the previous findings on page 24 (see also page 25). Moreover, given that independence and interdependence are multifaceted, we clarified our argument that to see a possibility that other factors related to culturally sanctioned ways of self and relationships can also contribute to cultural differences in social support seeking, we explore the role of empathic concern, which differs reflecting cultural norms regarding emotional expressivity and cognitive styles (pages 5-7). Furthermore, we referred to Miller et al. (2017) to address a possibility that norms about relationships reflecting an interdependent view of self and emphasis on either communal norms or exchange norms interact to affect hesitation and discomfort in social support seeking (page 28), because we think that future work addressing it is crucial to gain more integrated understanding about the cultural underpinnings of using social support. Finally, we also referred to potential effects of regional variations and socio-ecological variations, which can lead to differences in the extent to which individuals adhere to the dominant value of interdependence on page 28. 

Reviewer2

1) First, it is important to be specific about the cultures being investigated. The literature review moves back and forth between different conceptualizations, the connections among which are not clear. For example, are the phenomena in question specific to Japan? To all East Asian countries? Do they also include those in other countries of East Asian descent, such as Chinese Americans? Similarly, the comparison groups vary widely, including Canadians of European descent, Americans of British descent, etc. (The latter is not the majority of U.S. citizens, so this comparison is not particularly relevant to understanding country level differences in help-seeking.) Care should also be taken not to use "Canadians" or "Americans" when what is meant is a subgroup of the population.

Following this suggestion, we revised the framing of the introduction section to focus on East Asian culture and Western culture. And we also added some sentences to make the connection between different conceptualization clearer. Second, the reviewer pointed out that European Americans cannot represent “Americans.” Following this, we replaced the terms of “Canadians” and “Americans” with “European Canadians” and “European Americans” respectively.

2) The underlying arguments regarding culture and empathic concern are quite interesting, and as the authors note, some of the prior findings would appear on the surface to be contradictory. However, the line of argument is likely to be hard to follow without a more general introduction to the cultural attributes of the cultures under comparison (e.g., individualism/collectivism, self-concept, etc.). This could make subsequent discussion of possible reasons for cultural differences in help seeking easier to follow, particularly for readers who do not work in this area of research.

You pointed out that the arguments regarding cultural variations in empathic concern may be hard to follow without a more general introduction to the cultural attributes of the cultures under comparison. Following this suggestion, we added some sentences (line 55-64 on page 3-4) to introduce the distinct attributes of social relationship in each culture to help understand the cultural variations in relational concern. And we also described the attributes of each culture in emotional expressivity and cognitive thinking styles briefly at the beginning of 8th (line 122-125 on page 6) and 9th paragraph (line 131-134 page 7).

3) The paper would also be easier to follow if all key terms and phrases were defined on first use, including "empathic concern" in the first paragraph and abstract; "implicit social support" in paragraph 2, and so on.

Following this suggestion, we added the definition of empathic concern in the abstract (the 3rd sentence, line 23 on page 2). We deleted “implicit social support” in this revised manuscript. And the definition of social support was moved to the first sentence in the first paragraph (line 37-38 on page 3). The definition of loneliness is on line 154-155 (page 8).

4) My main concern is that comparing surveys distributed to different cultural groups is tricky and the authors do not seem to have considered cultural differences in the ways in which people respond to subjective scales. There is a large literature showing that simply translating a scale between languages does not ensure that the respondents are conceptualizing the different values on the scale in the same way. There are also related issues pertaining to social desirability in responses, which also varies across cultures.

As you pointed out, researchers have proposed several methods to improve the comparability of cross-cultural questionnaires. However, those statistical methods cannot effectively solve the major problems of cross-cultural comparison on Likert scales that Reviewer 1 mentioned (i.e., the reference group effect). There is only one way to address this issue that is to examine these findings with multiple methods in future. Therefore, combined with Reviewer 1’s last comment, we extended our argument on page 28.

---

## [Decision Letter · Decision Letter 1]

5 Jul 2021

PONE-D-20-40124R1

Cultural Differences in Social Support Seeking: The Mediating Role of Empathic Concern

PLOS ONE

Dear Dr. Zheng,

Thank you for submitting your manuscript to PLOS ONE. After careful consideration, we feel that it has merit but does not fully meet PLOS ONE’s publication criteria as it currently stands. Therefore, we invite you to submit a revised version of the manuscript that addresses the points raised during the review process.

Your paper has been reviewed for a second time. Although the changes and amendments done in your previous round of revisions seem partially suitable, the reviewer asks for a second set of revisions from you, in consideration of all the comments appended in the PDF that you will find in the attachment of this message.

We look forward to receiving your revised manuscript.

Kind regards,

Sergio A. Useche, Ph.D.

Academic Editor

PLOS ONE

Reviewers' comments:

Reviewer's Responses to Questions

**Comments to the Author**

1. If the authors have adequately addressed your comments raised in a previous round of review and you feel that this manuscript is now acceptable for publication, you may indicate that here to bypass the “Comments to the Author” section, enter your conflict of interest statement in the “Confidential to Editor” section, and submit your "Accept" recommendation.

Reviewer #1: All comments have been addressed

2. Is the manuscript technically sound, and do the data support the conclusions?

Reviewer #1: No

3. Has the statistical analysis been performed appropriately and rigorously? 

Reviewer #1: I Don't Know

4. Have the authors made all data underlying the findings in their manuscript fully available?

Reviewer #1: Yes

5. Is the manuscript presented in an intelligible fashion and written in standard English?

Reviewer #1: Yes

6. Review Comments to the Author

Reviewer #1: See attached PDF. I don't understand why this box requires a minimum of 100 characters. I don't understand why this box requires a minimum of 100 characters. I don't understand why this box requires a minimum of 100 characters. I don't understand why this box requires a minimum of 100 characters. I don't understand why this box requires a minimum of 100 characters. I don't understand why this box requires a minimum of 100 characters. I don't understand why this box requires a minimum of 100 characters. I don't understand why this box requires a minimum of 100 characters. I don't understand why this box requires a minimum of 100 characters.

7. PLOS authors have the option to publish the peer review history of their article (what does this mean?). If published, this will include your full peer review and any attached files.

Reviewer #1: **Yes: **Alan Page Fiske

---

## [Author Response · Author response to Decision Letter 1]

7 Sep 2021

Reviewer #1

This revision is responsive to many of the issues raised in the review of the original submission, and hence is much improved. However, the two most fundamental issues are not resolved. Since both of these issues are common to most research in cultural psychology, perhaps the best course of action is to acknowledge that them frankly from the beginning, and simply publish the paper. But I must say that what I see as the fundamental validity problems of most research in cultural psychology are salient in this paper. 

The issue of cross-national comparison of Likert scale values is frankly, though all too briefly, addressed in the next to last paragraph, though I suggest that the authors add a few sentences about better methods.

Thank you for the comments that shed light on the issues our manuscript could not handle. Following this suggestion by Reviewer 1, we revised the paragraph by adding sentences to raise the issue of the usage of self-report Likert scale and discuss the details (pages 30-31, lines 625-646). 

However, the authors continue to rely on the individualism – collectivism dichotomy, despite all the evidence that these are invalid constructs, that there is too much heterogeneity within nations to allow inferences as the national level, that different measures of the do not agree, that measures of them find dimensions rather than distinct categories, and that most studies that measure both constructs find little correlation between them (so they are not opposite ends of one dimension). Why do the authors base their metatheory on these invalidated constructs? My opinion is that they should at least acknowledge the many respects in which the constructs are problematic. However, I can also accept an editorial decision that doing so is beyond the scope of this paper. Even if this is the decision, however, there are component problems that I favor addressing. 

Could the starting point for the whole paper be wholly or partially a methodological artifact? The authors indicate that East Asian cultures (all? some? there are a great many East Asian cultures) are characterized by the judgment that presenting distress to another person typically(?) is an unwarranted or at least unwelcome imposition. If so, then one would expect social desirability effects such that survey respondents would under-report support-seeking. This likelihood needs to be carefully addressed.

Following this suggestion by Reviewer 1, in the second half of the introduction section, addressing the issues of the usage of self-report Likert scale and the West-East dichotomy, we stated that this research, which uses self-report scales and relies on the West-East dichotomy, has theoretical and methodological flaws in the context of cultural psychology work (pages 3-4, lines 35-55). Additionally, in this research, we failed to address potential differences within East Asian cultures. We added sentences to mention that by adopting the method of triangulation to identify explanatory factors of cultures by comparing subgroups, researchers can provide an advanced view of how the mind is shaped by cultural content beyond the dichotomy of individualism and collectivism (pages 29-30, lines 610-616). Furthermore, we also addressed the issue of social desirability in details (page 31, lines 637-646).

Let me return to the issue of comparing distributions of responses on items in different languages. In the wording of a question, even small differences within a language may yield large quantitative differences in responses. But here we are comparing responses to items in different languages. The differences in social support-seeking in these studies and those reported in previous studies are based on comparing scales anchored by English words such as rarely, very much, and so forth, with scales in Mandarin, Cantonese, Japanese, Malay, Vietnamese, Thai, or Korean lexemes with similarly indeterminate meanings that surely do not align perfectly with the English frequency terms anchoring the scales for the North Americans. Back-translation is a start, but what one inevitably discovers doing back translation is that it is just not possible to “ensure cross-cultural equivalence” of the purportedly corresponding items in two languages. Hence differences between nations in the distributions of responses cannot be used to make valid inferences about the difference in the absolute distribution of responses in the samples (including their means).

Furthermore, the lexemes used to describe forms of “social support-seeking”, or the term for “stress”, or the term “solvable” or “responsible”, or the lexeme for “close to” in a scale in English cannot possibly have denotations (nor connotative valuation) identical to the lexemes used to describe those constructs in any other language – here, Japanese. Languages simply do not have lexicons that that construe the world according to any universal ontology or taxonomy: lexicons do not align in one-to-one correspondence.

As Reviewer 1 mentioned, the effects of culturally biased ideas participants rely on are inescapable regardless of the careful implementation of translation and back-translation, which we stated on page 30. Following your suggestion, we deleted the phrase “ensure cross-cultural equivalence,” which had been originally used to explain the scale of relational concern in Study 2.

In addition, in Study 2 the authors found substantial differences in the problems for which US respondents potentially sought social support, compared to the Japanese respondents; in particular, the Japanese respondents reported more negative and more stressful events (though one does not know whether these are true differences, or result from differences in the meanings of the scales in the respective languages). Given that finding, how meaningful is it to compare the social support-seeking of the two samples? Surely the nature of the problems one faces has a large effect on whether one seeks support, what support one seeks, and from whom. 

As Reviewer 1 pointed out, in Study 2, compared to American participants, Japanese participants perceived the stressful events they described as more stressful, negative, and controllable and felt more responsible for the events. Although we controlled the levels of participants’ feelings related to the events in a series of the multiple mediation analyses, the unexpected differences in the feelings across cultures suggest that a follow-up study is warranted for the examination of the associations among empathic concern/relational concern, support seeking, and loneliness in a more controlled setting—such as a hypothetical vignette including a commonly experienced stressful event. In this revision, we mentioned the limitation of Study 2 (pages 24-25, lines 496-502).

Beyond that, the words, gestures, facial expressions and other actions that comprise “support-seeking” in Japan are doubtless different from those that are typical in Canada, or in the US. That is, the very acts of support-seeking differ between the two nations, meaning that the studies compare apples in North America to oranges in Japan. Furthermore, the “support” that is sought surely differs across cultures, nations, and regions. For example, Adams & Plaut (2003) found that, in contrast to European Americans, West Africans expect financial support from friends, but rarely seek advice and even less often expect sympathetic listening to expressed emotions. (My own informal observations in the course of over 4 years in West Africa support this.) So, in sum, just what were the Japanese respondents reporting that they were seeking, and how were they seeking it? What were the Canadian and what were the US respondents seeking (not necessarily the same thing, as the two nations contain somewhat different sets of cultures), and what did seeking consist of? 

All in all, I believe that the authors need to assiduously unpack and discuss the meanings of the prompts and of the scale anchors in the respective languages. A Likert self-report scale is not a caliper that can be reliably calibrated so as to yield objective and truly comparable measurements, and Likert self-report scales in (very) different languages simply do not measure the exact same behaviors, let alone absolute frequency distributions of behaviors in any of the samples. In short, can one truly conclude that there are, in fact, between-nation differences in the rates of behaviors related to the construct of “social support-seeking” – especially if people in the respective nations are not seeking exactly the same sort of support in the same way?

As Reviewer 1 pointed out, this research, which relies on self-report Likert scales, failed to address what support people seek. Although this research focused on explicit forms of social support seeking such as getting emotional support and advice from other people, people may seek more implicit forms of support defined as the emotional comfort experienced without disclosing one’s problems and stress. Previous research suggests that Asians and Asian Americans likely benefit from implicit support seeking, and those who tend to endorse adjustment goals are likely to emphasize relational concern as a motivating factor in deciding to seek implicit social support. Given the previous findings, it is not clear whether both empathic concern and relational concern are associated with not only explicit forms of social support seeking, but also implicit forms of social support seeking. As we added sentences on pages 27-28 (lines 559-573), future work will need to address the possibility that the mediating roles of empathic concern and relational concern would depend on the forms of social support seeking.

In any case, I do not have great confidence in the validity of claims such as, “compared to Americans, Japanese and Chinese individuals reported greater degrees of loneliness”. And if Japanese are more lonely, how does that comport with the notion that Japanese culture is more ‘collectivist’ than North America? Likewise, I doubt the validity of the evidence that, “compared to East Asians, Westerners are more likely to empathize with people in distress by exhibiting sympathy (e.g., [27]). For instance, Cassels et al. [28] found that East Asian adolescents reported lower empathic concern than did Western adolescents, whereas mainland Chinese university students scored lower in empathic concern assessments than German undergraduates [29, 30]. Moreover, American counselor trainees showed greater empathic concern than their Thai counterparts [31] . . . . East Asians show less empathic concern than do Westerners.” Simply running Student’s t-tests and ANOVAs on Likert scales in two different languages does not constitute evidence for such conclusions. The authors have added a short penultimate paragraph to acknowledge this, but then, why do they make the inferences they do make throughout the paper?

In this revision, we emphasized that this research mainly aimed to describe whether, and to what extent, cultures influence relationships among a set of variables and what factors account for the cultural differences—rather than just reporting what individuals think self-reflectingly about themselves (on page 4). To do so, we revised the abstract and the title of the section (page 6, lines 100-101) and added some sentences in the main text (page 3, lines 31-34, page 5, lines 63-68, page 6, lines 90-94, and page 10, lines 192-193). As Reviewer 1 pointed out repeatedly, we assessed cultural differences in social support seeking, empathic concern, relational concern, and loneliness in an inappropriate manner, and we admit the methodological flaw. To figure out the relationships among these variables, however, we firstly have to show these cultural differences based on the results of t-tests. Thus, while we still report cultural differences in social support seeking, empathic concern, relational concern, and loneliness in this revision, we tried to avoid phrases to emphasize these cultural differences (e.g., page 10, lines 178-179 and 183-184) and draw attention to the usage of self-report Likert scale when mentioning the previous findings (page 7, lines 125-129). 

Again, let me say that I realize that this critique applies to most research in the cross-cultural psychology paradigm. Yes. 

These issues are probably somewhat less vitiating for mediational analyses (so long as one does not compare the magnitudes of mediations in samples studied with instruments in different languages). Hence there remains a lot of value in these studies. I would suggest that the authors could greatly strengthen the interesting but unsupported inference that people (in general) are less disposed to seek social support when they view support-seeking as an imposition on relationships, and hence potentially deleterious to them. To explore this, why not ask informants if this is indeed what they think? It would be very illuminating to do 20-30 interviews and a couple of focus groups in Japan, and the same in North America. My intuition is that, for the most part, North American recipients of requests for social support feel that such requests are flattering, even moving, indicators of trust and closeness. I believe that in my own North American subculture, support-seeking strengthens friendships and family bonds.

Due to the limited time in revision and the current situation of COVID-19, we could not conduct an interview survey that Reviewer 1 proposed. However, Miller et al. (2017) conducted interviews and tested Indian, Japanese, and North American participants by utilizing a hypothetical situation in which a person needs help and asking questions about reliance on exchange norms, relationship maintenance concerns, and social support (e.g., comfort in asking for help). They demonstrated that Indians were less likely to endorse exchange norms than Japanese and North Americans, and that the cultural difference in exchange norms accounted for more positive social support outlooks in Indians than Japanese and North Americans. Additionally, when comparing the Japanese and North Americans, relationship maintenance concerns mediated the cultural differences in social support. In this revision, we mentioned the results (pages 5-6, lines 82-89). 

What is the evidence that East Asians “are less likely to experience emotion more intensively”? That claim astonishes me. And how does this mesh, for example, with the fact that suicide rates in Japan are nearly twice as high as in, say, the United Kingdom? Or with, say, the Japanese preoccupation with kawaii? Suicide and kawaii-related practices manifest themselves quite publicly, so how does this comport with the claim that East Asians are less likely to “express their emotion publicly”? At baseball games?

We revised sentences referring to emotional suppression and expressivity (page 8, lines 145-147). 

How were non-European participants excluded from the Alberta sample – and what definition of “European Canadian” was used to do so? Likewise for study 2, how were non-European Americans defined and how were they excluded?

To explain the detail, we added sentences “The Canadian participants were prescreened based on their self-defined ethnicity” in Study 1 (page 11, lines 217-218) and “The American participants were recruited with filters on self-defined ethnicity (European American) and nationality (American)” in Study 2 (page 16, lines 320-321).

In the general discussion, the authors write that “individuals from Japan display less empathic concern for people in distress. These cultural differences result from the cultural differences in emotional expressivity and cognitive thinking styles. On the one hand, East Asians, with their strong motivation to maintain interpersonal harmony, compared to Westerners, are more cautious about their emotional responses in interpersonal contexts (e.g., [10]). The high tendency to restrain and suppress emotion in daily life may hinder East Asians from empathizing with unfortunate others”. As far as I can see, their studies do not address the claims that I underline, nor do the authors offer any other evidence for these causal inferences. (And what does “more cautious” mean?) Is this claim about all emotions?

As Reviewer 1 pointed out, our argument regarding emotion suppression and expressivity was not appropriate in this context. We thus deleted it in this revision.

The authors note that “compared to Americans, Chinese people are more likely to perceive unfortunates as responsible for and deserving of their misfortune, and they reported less overall sympathy [41]. In Study 2, we also consistently found that, compared to European Americans, Japanese people reported more responsibilities for their own stressors”. To me these results seem completely at odds with the characterization of Chinese or Japanese as ‘collectivist’ and European Americans as ‘individualist’. Individualists should believe that people are individually responsible for what happens to them, while collectivists should believe that misfortunes and stressors are due to the overall circumstances, roles and relationships, not the individual, personally. Indeed, Joan Miller found precisely this contrast between explanations for real events offered by her samples in Chicago and Orissa, India (sorry, I can’t recall the reference – from the 1970’s, I believe). I find that I don’t understand what construct of ‘collectivism’ the authors are using.

As Reviewer 1 pointed out, we admit that this argument on the perception of responsibilities was insufficient. We thus deleted it in this revision.

It seems to me that “turning to others in times of need helps restore one’s sense of belonging” would be especially true, or at least especially prevalent, among ‘collectivists. In any case, if turning to others in times of need helps restore one’s sense of “belonging,” how does belonging differ from a “relationship”? Why does such support-seeking strengthen belonging but threaten relationships? 

I reiterate that the authors of this paper operate in a paradigm that I find problematic – I have essentially the same problems with nearly all research in cultural psychology. So an editorial decision to put aside my critique would be reasonable. On the other hand, paradigms only change when they are challenged study by study, method by method, findings by findings.

Given the Reviewer 1’s concern, we deleted the phrase “turning to others in times of need helps restore one’s sense of belonging” in this revision.

Again, thank you for the valuable comments and providing us with a chance to revise our manuscript along them. 

Reviewer #2

While this revision fixes minor issues raised in prior reviews, the authors have not done any new analyses to address the problem of cultural differences in scale use, and the results are still stated as if differences on these scales correspond to actual cultural differences. Yet as both R1 and I mentioned in the first round of reviews, there is a lot of evidence that this is not the case. There are techniques the authors could have used to examine this issue (e.g., centering). The results may or may not have been the same. But since they did not conduct such analyses and their explanation in the cover letter for this is not compelling, my recommendation is to reject the paper.

We apologize for our mistake that we failed to address the last point raised by Reviewer 2 in the previous review. In this review, we confirmed that we found the same results even if the scores of all related variables were centering. In this revision, we mentioned explicitly that the scores of all related variables were centering before the mediation analyses (page 13, lines 256-257 and page 19, lines 394-395).

---

## [Editor Report · Decision Letter 2]

16 Dec 2021

Cultural Differences in Social Support Seeking: The Mediating Role of Empathic Concern

PONE-D-20-40124R2

Dear Dr. Zheng,

We’re pleased to inform you that your manuscript has been judged scientifically suitable for publication and will be formally accepted for publication once it meets all outstanding technical requirements.

Kind regards,

Sergio A. Useche, Ph.D.

Academic Editor

PLOS ONE

Additional Editor Comments (optional):

Thanks for the revisions made to the paper. After a careful review, I believe the authors addressed well the remaining comments. Therefore, the paper can be considered as publishable in PLOS ONE.

---

## [Editor Report · Acceptance letter]

19 Dec 2021

PONE-D-20-40124R2 

Cultural Differences in Social Support Seeking: The Mediating Role of Empathic Concern 

Dear Dr. Zheng:

I'm pleased to inform you that your manuscript has been deemed suitable for publication in PLOS ONE. Congratulations! Your manuscript is now with our production department. 

Kind regards, 

on behalf of

Dr. Sergio A. Useche 

Academic Editor

PLOS ONE